# Comparative Evaluation of *Piper nigrum*, *Rosmarinus officinalis, Cymbopogon citratus* and *Juniperus communis* L. Essential Oils of Different Origin as Functional Antimicrobials in Foods

**DOI:** 10.3390/foods9020141

**Published:** 2020-01-31

**Authors:** Katarzyna Leja, Małgorzata Majcher, Wojciech Juzwa, Katarzyna Czaczyk, Marcin Komosa

**Affiliations:** 1Department of Biotechnology and Food Microbiology, Faculty of Food Science, Poznań University of Life Sciences, Wojska Polskiego 48, 60-627 Poznan, Poland; wojciech.juzwa@up.poznan.pl (W.J.); katarzyna.czaczyk@up.poznan.pl (K.C.); 2Institute of Food Technology of Plant Origin, Faculty of Food Science, Poznań University of Life Sciences, Wojska Polskiego 31, 60-624 Poznań, Poland; malgorzata.majcher@up.poznan.pl; 3Department of Animal Anatomy, Faculty of Veterinary Medicine and Animal Sciences, Poznan University of Life Sciences, Wojska Polskiego 71c, 60-625 Poznań, Poland; marcin.komosa@up.poznan.pl

**Keywords:** antibacterial mode of action, bacteriostatic activity, essential oils, flow cytometry, viable but nonculturable cells

## Abstract

Essential oils can be used as preservatives in foods because of their ability to inhibit bacteria growth in low concentration, which does not influence on foods’ organoleptic properties and does not generate the resistance mechanisms in cells. The aim of that work was to compare the effectiveness of commercial oils from black pepper (*Piper nigrum*), rosemary (*Rosmarinus officinalis*), lemongrass (*Cymbopogon citratus*) and juniper (*Juniperus communis* L.) with oils obtained in our laboratory. The typical cultivation method was supported by the flow cytometry to detect the cells of very low physiologic and metabolic activity. Our investigation demonstrated that both types of oils can effectively inhibit the growth of saprophytic bacteria *P. orientalis*. The oils distilled in our laboratory had a bacteriostatic effect at a lower concentration, which is important for application in the food industry. Flow cytometry analyzes and confirmed the thesis that essential oils do not have a germicidal effect on bacteria cells.

## 1. Introduction

Plants are a source of many compounds not only nutritious (such as proteins starches or fats [1,2,3,4,5,6] or antimicrobial, among which essential oils play an important role. Plant essential oils (EOs) are a natural blend of various organic compounds that are characterized by a strong fragrance. EOs are synthesized in oilseed plants during secondary metabolism. They are transparent and well soluble in both organic and lipids solvents. Since they are a natural plant product, their chemical composition can vary depending on climatic conditions, where the plant was cultivated, and is different in various parts of the plant from which the oil was obtained. The content of bioactive compounds in EOs is also dependent on the extraction method applied in the process [7,8].

Oilseed plants are very promising alternatives for commonly used food preservatives because of the antibacterial activity of their bioactive compounds (such as eugenol, citral, pinene, myrcene, linalool etc.) [9,10]. Plant EOs also have other confirmed beneficial biological effects that make them attractive for the food industry, such as appetite stimulation, immunomodulatory and antioxidant effects as well as antiparasitic, anaesthetic and antistress activities [11]. Thus, essential oils have the potential to become a new generation of food products for people and animals, replacing chemical preservatives in foods, and antibiotic growth promoters in animal diets [7,8]. One of the main advantages of replacing the chemical preservatives in foods by EOs is its documented lack of bactericidal activity (in low concentrations). It means that bacteria in contact with an appropriately selected concentration of oils does not trigger resistance mechanisms in cells, thanks to which in the future there will be no problem of immunizing bacteria against bioactive compounds contained in oils [12]. That is why determining the lowest bacteriostatic concentration of oils (so-called MIC value—Minimal Inhibitory Concentration) is so important. For example, Burt et al. [12] appointed the MIC values of selected EOs concentration for some gram(−) bacteria (Listeria monocytogenes, Salmonella typhimurium, Escherichia coli O157:H7, Shigella dysenteria, Bacillus cereus, Staphylococcus aureus) at a level between 0.2–10 µL/mL.

They also underline very important issues in food products with higher values, which are estimated in vitro. Their research has shown, that in fresh meat, meat products, fish, milk, dairy products, and vegetables, values increase to 0.5–20 µL/g.

There are many papers on the antimicrobial effect of commercial EOs, e.g., [13,14,15], as well as fresh EOs pressed from plants [12,16,17]. For example, Burt et al. [12] investigated the antibacterial activity of oregano and thyme oils against *E. coli* O157:H7 by disc diffusion essay as well as by microdilution colorimetric assays. They concluded that both oils in low concentrations may be effective in reducing the number or preventing the growth of *E. coli* O157:H7 in foods. Sacchetii et al. [16] tested basil and thyme oils against *Staphylococcus aureus*, *Enterococcus foecalis*, *Pseudomonas aeruginosa* and *Escherichia coli*. The antibacterial activity was estimated only by disc diffusion method and MIC values were calculated. As a result, it was stated that all the tested strains were inhibited to some degree so that oils could be used as a natural preservative in foods. Therefore, as we can see, commonly, only typical cultivation methods to estimate the antibacterial efficiency of plant materials are described [12,16].

In the first step of our research, we investigated the antibacterial mode of action of nine commercial oils [18]. The four oils that have the most effective antibacterial activity against investigated *P. orientalis* strains (pepper, rosemary, lemongrass and juniper oils) has been chosen for further research. The aim of that work was to compare the chemical composition of both types of above-mentioned oils and the efficiency of their antibacterial activity. In that aim, the complete analytic protocol (including both cultivation and alternative methods) developed by us was used.

## 2. Materials and Methods

### 2.1. Bacteria Strains

Two *Pseudomonas orientalis* strains designed as P49 and P110 from the strain collection at the Department of Biotechnology and Food Microbiology, Poznan University of Life Sciences (Poland) were used in the study. Stock cultures were kept on enriched broth medium (BTL, Lodz, Poland) at 4 °C and renewed bimonthly.

### 2.2. Essential Oils

Plant material was purchased from Eco Farma (Łódź, Poland), a company with a certificate of authenticity and all the necessary quality certificates. EOs were extracted from black pepper (*Piper nigrum*) fruits, rosemary (*Rosmarinus officinalis*) leaves, lemongrass (*Cymbopogon citratus*) leaves and juniper (*Juniperus communis*) fruits by hydro-distillation using a Clevenger-type apparatus. In total, 100 g of dried powdered material of each plant was placed in the 1 L round-bottom flask with 300 mL of distilled water and boiled for 2 h. The plant oils were collected, dried over anhydrous sodium sulfate and stored in darkness in hermetic vials at 4 °C before use.

Commercial equivalents of tested oils were obtained from Company from Poland. The essential oil samples were stored in glass vials with teflon-sealed caps at 4 °C in the absence of light.

### 2.3. Analysis of Bioactive Compound of Investigated Essential Oils by Gas Chromatography (GC) Technique

The GC-MS analysis were performed using a Hewlett-Packard HP 7890A GC coupled to a 5975C MS (Agilent Technologies, SC, USA) with a SLB-5ms column (Supelco) (30 m × 0.25 mm × 0.5 µm). Operating conditions for GC/MS analysis were as follows: helium flow, 0.8 mL/min; initial oven temperature, 40 °C (2 min), raised to 240 °C at 8 °C/min rate, held for 6 min isothermally. For all peaks retention indices were calculated to compare results obtained by GC/MS with literature data and mass spectra of eluting compounds to those of the NIST 05 library match (NIST MS Search v2.0, Toronto, Canada 2005). Retention indices were calculated for each compound using homologous series of C7–C20 n-alkanes. Mass spectra were recorded in an electron impact mode (70 eV) in a scan range of m/z 33–350 [18]. The ion source temperature was set at 200 °C. The composition of EO has been expressed as the percentage composition calculated from the chromatogram obtained on the SLB-5 column. Normalized peak area % were calculated based on the total ion chromatogram (TIC) without obtaining response factor for particular compounds.

### 2.4. Antibacterial Activity of Tested EOs

The biological activity of EOs against *P. orientalis* strains was determined by employing the standard discs diffusion technique. In total, 150 mL of Mueller-Hinton medium was poured into Petri dish on a horizontal surface to give a depth of 3 mm. The plate was seeded with an overnight Enriched-Broth culture adjusted to approximately 10^5^ CFU/mL by the agar overlay method. After the agar had set the seeded plates were dried for 30 min at 36 °C and then the discs were applied. On one plate five discs could be accommodated without confluence of zones. Four discs contained individual EO and one contained gentamicin—25 µg/disc (a control probe). The EOs were dissolved in 10% aqueous dimethylsulfoxide (DMSO) with Tween 80 (0.5% *v*/*v*) and sterilized by filtration through a 0.45 μm membrane filter. Sterilized discs (Whatman no. 5, 6 mm diameter) were impregnated with the respective EOs (25 µg/disc) and placed on the agar surface. A standard disc containing gentamycin (25 μg/disc) was used as reference control. All Petri dishes were sealed with sterile laboratory parafilm to avoid evaporation of the test samples. Plates were incubated at 36 °C for 18 h and zones were measured with sliding calipers and dark-ground illumination. Studies were performed in duplicate. In the next step the minimal inhibitory concentrations of investigated EOs were determined. For that aim mother cultures of both *P. orientalis* strains were set up 24 h before the assays in order to reach the stationary phase of growth. Subsequently, the Minimum Inhibitory Concentrations (MIC) of oils were determined by the method recommended by the National Committee for Clinical Laboratory Standards [12] with some minor modifications. To enhance EOs solubility the Tween-20 (Sigma) was incorporated into the Enriched-Broth medium with agar in a final concentration of 0.5% (*v*/*v*). A series of two fold dilution of each EO, ranging (determined based on previous observations) from 6.0 to 96.0 mg/mL, was prepared in Muellur Hinton agar at 48 °C. Petri plates were dried at room temperature for 30 min prior to spot inoculation with 3 μL aliquots of culture containing approximately 10^5^ CFU/mL of each *P. orientalis* strain. Then, cultures were incubated (36 °C, 18 h) and MIC values were determined. Inhibition of bacterial growth in the cultures containing EO were judged by comparison with growth in control plates without EO. Experiments were carried out in duplicate and the difference between the value of duplicates tests are presented. The MIC value is the lowest concentration of EO able to inhibit visible growth of tested bacteria. These observations were confirmed in spectrophotometric analysis. It enabled the determination of the bacterial growth kinetics in the optimal conditions and the combination with the kinetics of growth inhibition in the presence of the essential oil (based on the bacteria number as log CFU/mL, the percentage of growth inhibition was calculated).

### 2.5. Estimation of Intracellular Metabolic Activity of P. Orientalis by Flow Cytometry

The intracellular metabolic activity of *P. orientalis* P49 and P110 strains was investigated by measurement of the redox potential using a flow cytometer equipped with four lasers (375, 405, 488, and 633 nm), 11 fluorescence detectors, forward scatter (FSC) and side scatter (SSC) detectors (BD FACS Aria Aria™ III, Becton Dickinson, NJ, USA). In total, 1 mL of Enriched-Broth Medium was inoculated with bacterial suspension (10% vol/vol) and supplemented with tested essential oil in MIC concentration. A control sample was similarly prepared, but no essential oil was added. Cytometric analysis was carried out after 10 min of incubation of bacteria with oils and after 24 h and 72 h of incubation (at 15 °C). For that aim, the oil phase was removed and the probes were centrifuged (5000 rpm, 5 min), the bacterial pellet was diluted with 250 μL of PBS. Cells were stained with 1.5 µL RedoxSensorTM Green and 0.7 μL propidium iodide. After 10 min of incubation without light, the analyses were done. Each sample was analyzed in triplicate. The estimation of cells redox potential was performed using medians of green fluorescence signals of gated populations defined on bivariate dot plot [18].

### 2.6. The Integrity of Bacteria Cell Membrane by Determining Protein Content by the BCA Method

The concentration of protein released into the cell suspension due to the action of essential oils was determined by the bicinchoninic acid (BCA) method. To prepare the reaction mixture, 1 mL of the reagent for protein assay was mixed (1 mL of 4% CuSO₄·5H₂O + 49 mL of BCA solution), 50 μL of *P. orientalis* culture with the addition of essential oil (10% volume) and 1 mL of liquid broth were mixed and incubated at 60 °C for 15 min under continuous shaking. Then, the samples were cooled at room temperature and subjected to spectrophotometric analysis using a Specord 205 UV/Vis spectrophotometer (analyst Jena, Germany). Absorbance was measured at wavelength λ = 562 nm. A control sample was prepared analogously, but with no addition of an oil. A blind attempt was a liquid broth. To determine the protein concentration in the sample, a calibration curve was prepared using standard solutions of bovine serum albumin (BSA) in the concentration range of 0.05—1 mg/mL.

### 2.7. Observation of the Shape of Bacterial Cells with the Inverted Microscope

The inverted microscope was used to observe the actual shape of *P. orientalis* cells and changes in cell morphology under the influence of oils. The inverted microscope enables observation of the specimen from the bottom. For this purpose, the tip of the lens is pointing upwards. The objective is under the stage and light is directed on the specimen from above. To obtain the preparation, 1 mL of the bacterial culture was centrifuged (3000 rpm, 5 min). The bacterial pellet was smeared on a glass slide. After drying, the smear was stained with crystal violet. Preparations were observed using an immersion lens (100×). The cell length and width were measured by Axiovert 200 program.

## 3. Results and Discussion

In our work, the mechanism and efficiency of antibacterial mode of action of commercial and our own production plant EOs were compared. The commercial oils were received from the Company and were produced by a hydro-distillation method. Oils produced in our laboratory were obtained from the plant material purchased in Eko Farm (Łódź, Poland) and some material was also obtained by a hydro-distillation method. In our previous work [19], we described a preliminary study on antibacterial effects of commercial EOs. Since the results showed an antibacterial potential of the investigated EOs, we decided to continue our study and compare some commercial oils with the ones that were produced by us. In fact, production of oils from plant sources is a very cumbersome, long-lasting and slow process. Table 1 presents efficiency of obtaining oils from plant raw materials and includes a comparison of the cost of 10 mL of our own oils and the oils that we purchased for this study.

It occurred that the cost of black pepper and juniper oils is nearly the same for both commercial oils and those created in our laboratories. The cost of production of rosemary and lemongrass oils is almost three times higher compared to commercial counterparts, which results from low efficiency of their production.

As one can see, the prices of ready-made essential oils and the plant materials are comparable. Thus, obtaining oils by ourselves is more expensive (the cost of equipment, reagents and human work were not included). In the next step, the bioactive composition of both types of oils was tested. To compare the content of bioactive components, a GC/MS analysis was performed. The results are summarized in Table 2.

In the case of most components, the percentage of bioactive components is comparable. Significant differences were observed in the case of juniper oil. The LJN oil has a higher concentration of α-pinen in comparison with the commercial one (33.1% contra 1.0%), sabinene (5.21%, in the commercial oil this compound was not detected), β-myrcene (11.8% contra 1%), γ-terpinene (5.5%, not detected in commercial oil), β-caryophyllene (8.5%, not detected in commercial oil). Only a higher concentration of β-pinene and citral in commercial oil was detected (accordingly, 18% and 4.3% contra 12.2% of β-pinene and no citral was detected in the oil obtained by the authors). In CBP oil sabinene, α-phellandrene and γ-terpinene were not detected, but in LBP oil a concentration of sabinene was on the level of 10.8%, of α-phellandrene was equal 4.6%, and γ-terpinene was 15.8%. In the case of linalyl acetate and delta-3-carene the situation was reversed—they were detected only in CBP oil, accordingly on the level of 22% and 9%. The bioactive compositions of both rosemary oils were very close, with only CRS oil having a higher concentration of β-pinene (8% contra 1.6%), γ- terpinene (18% contra 1.0%), camphor (20% contra 9.5%). On the other hand, the concentration of 1.8-cineol was higher in LRS oil than in commercial one (44.2% contra 18%). In the LLG oil, 1.8-cineone (4.3%) and camphor (4.5%) were detected, while they were not observed in CLG one. In CLG oil, a high concentration of citral (67%) was observed, but citral, surprisingly, was detected in LLG oil in lower concentration (31.2%). Such differences in both oils’ composition certainly influenced the effectiveness of antibacterial action of essential oils, because every single compound has a strictly defined role in antimicrobial activity. Summarizing, the highest diversity in percentage composition of both oils was observed in the case of α-pinene, sabinene, myrcene, terpinene, caryophyllene, and citral. α-pinene is a major component of many bioactive plants, such as scots pine (*Pinus sylvestris*) and eucalyptus [18]. The antibacterial mode of action of α-pinene is based on their radical scavenging and superoxidase action, which inhibit respiration and ion transport processing (leaked potassium and phosphate); it also suppressed enterotoxins production, caused release of cellular content, morphological changes, permeabilization of membranes, and dissipations pH gradient [19,20,21].

Caryophyllene inhibits decarboxylase in *Enterobacter aerogenes*. In *Pseudomonas aeruginosa*, it causes depolarization and permeabilization of membrane, leaking and coagulation of cytoplasmic contents; it can also inhibit respiration activity. In *Staphylococcus aureus*, it entered a viable but non-cultivable state and lost membrane activity. Citral is a lemon scented acyclic monoterpene aldehyde, which consists of a racemic mixture of two isomers, geranial and neral. It is the most important component of essential oils of *Cymbopogon* species. A high concentration of that bioactive compound is also observed in the tropical plant *Backhousia citriodora* [22]. Its biostatic activity includes activities such as inducing free radicals formation in bacteria cells and oxidative damage to bacteria DNA. It also damages cell membranes [23]. In the next step, we tried to determine zone inhibitions, MIC values and the percentage of bacteria growth inhibitions for both types of oils, and later we compared the results obtained, as seen in Table 3.

The only exception was lemongrass oil (in the case of P110 strain), for which brightness zones were very similar, regardless of the origin of the oil. Generally, the zone of growth inhibition was significantly higher for the P110 strain than that for P49. This fact is likely linked with the differences in their cell wall composition. Our previous investigation, which is not yet published, demonstrated that isolation of DNA from *P. orientalis* P49 cells is very tedious and inefficient, while the *P.orientalis* P110 strain does not cause such difficulties. An analysis of the chemical composition of commercial oils and those extracted by us did not show very significant differences for the most of the ingredients. However, these differences significantly influenced the MIC value results. The MIC values were about 10 times lower for oils obtained in our laboratories than for purchased oils for both tested strains. For both strains, P49 and P110, the lowest MIC value was observed in the case of LLG oil. That MICs were ca. 3 d.m. mg/mL, and were under 20 times lower that the ones observed for commercial oils. Moreover, the percentage rate of inhibition of bacteria growth was higher in case of oils from our laboratory. The highest inhibition (88%) of P49 growth was observed in the case of LLG oil, while that of P110 was inhibited in the highest degree (88%) by LBP oil. These observations indicate that a synergistic interaction between individual bioactive components is a very important factor here. Probably, this effect is more significant in relation to antimicrobial activity than to the composition of individual phytochemicals. The synergic effect has been described in many papers. Among others, Turgis et al. [24] observed such a synergic effect of bacteriocins and essential oils. Several researchers stated that *Origanum vulgare* EO induced a synergistic effect against *L. monocytogenes*, whereas the combination of nisin with *Thymus vulgaris* EO caused a synergistic effect against *S. typhimurium*. Pediocin caused a synergistic effect against *E. coli* O157: H7 when it was combined with the *Satureja montagna* EO. Cassella et al. [25] confirmed the synergic effect of tea tree and lavender essential oils common causes of tinea infection in humans caused by *Trichophyton rubrum* and *T. mentagrophytes* var. *interdigitale*.

Both methods, disc diffusion and MIC estimation, are burdened with errors. The main disadvantage of the disc diffusion method is the fact that different compounds diffuse in the medium at different speeds. The method is not mechanized, so there is also a potential human error. In the MIC method, the effect is described based only on turbidity of the medium. It is not possible to determine whether viable but nonculturable cells [26,27] are present or not. Thus, results needed additional confirmation by determining bacterial growth curves both in the presence of oil and without oil (control test), as well as by flow cytometry analyses, which allow to divide cells into three groups depending on their activity (high activity, medium—viable but nonculturable cells and death cells) [28]. The growth curves are presented for most effective oils for P49 (juniper oil which reduced the bacteria cells about 88%) and P110 (black pepper oil which reduced about 80% of bacteria cells) strains (Figure 1).

The numbers of bacteria were analyzed through 10 h in cultivation, first without oil and next with commercial and laboratory oils at MIC levels. As can be seen, both oil types reduced the number of live bacteria cells, but the laboratory oil was more efficient. The LJN oil reduced the growth of *P. orientalis* P49 about 5 log cycles (CJN oil about 3 log cycles) and LBP oil reduced the growth of *P. orientalis* P110 about 6 log cycles (CJN oil about 4 log cycles). A similar observation was described by Moraes-Lovison et al. [29]. They presented the in vitro antimicrobial activity of oregano essential oil against two foodborne pathogens *S. aureus* and *E. coli*. The oregano oil in MIC level (0.6 mg/mL in case of *E. coli* and 0.56 mg/mL in case of *S. aureus*) significantly reduced the growth of bacteria after 72 h of cultivation (about 5 log cycles in *E. coli* and 3 log cycles in *S. aureus*). The results of flow cytometry analyses are presented in Table 4.

Additionally, two-dimensional graph of the distribution of individual fractions in cytometric analyses is presented on the example of pepper and lemongrasses oils (Figure 2).

The distribution of fractions by cell activity was significantly different for commercial oils and oils produced in laboratory conditions. The share of cells in the high-active fraction was many times lower in the case of oils extracted in our laboratory. This result confirms our previous observations: the MIC value of commercial oils was also significantly higher. The most spectacular effects were observed in the case of lemongrass oil (for both strains). The fraction of active cells was only 1.6% in case of P49 strain and 1.2% for P110 strain. Black pepper was very effective, too. The share of cells in the active fraction has been reduced to less than 30%. An important observation from these studies is a confirmation of the presence of viable but non-culturable cells—which cannot be observed in typical culture methods [30].

Essential oils destroy the wall and cell membrane of bacteria, causing the losses of vital intracellular materials, which is manifested by changes of the shape of the cells [31]. The results of such morphological transformations under the influence of both types of oils are presented in Table 5.

*P. orientalis* cells become round under the influence of both types of oils. Cell sizes are slightly different. Cells under the influence of oils tend to agglomerate. Bacteria were stained with crystal violet, but the color of the cells varies from one preparation to another. This is due to the fact that the oils are greasy and grease affects the penetration of the dye into the cell; it is also difficult to rinse the background well. Changes of bacteria cells’ size under the influence of essential oils were also observed by other scientists. For example, Zhang et al. [32] observed that cinnamon oil in MIC level (4.0 mg/mL in case of *E. coli* and 2.0 mg/ML in case of *S. aureus*) changes the morphology of bacteria cells indicating cell damage. Bouhdid et al. [33] confirmed that oregano oil is active against *S. aureus* and *P. aeruginosa* at MICs of 0.031% and 1% (*v*/*v*), respectively. In microscopic preparations, an accumulation of membranous structures in the cytoplasm and a significant amount of cytoplasmic material in the surrounding environment of the oil-treated cells were observed. The result of this was shape changes and reduction of bacterial cells size. Such changes in bacteria cells are strictly linked with releasing proteins from their wall. Therefore, the concentration of proteins (under the influence of essential oils in MIC level) in the cultivation environment was also determined. The results are presented in Figure 3.

The concentrations of proteins were highest in all cultivations with the addition of oils in MIC concentrations. The highest ratios were observed in oils produced in our laboratory. The most efficient effluence of proteins was observed in lemongrass and black pepper oils. These oils violate the integrity of the cell membrane and cause partial or complete leakage of cell contents (including proteins, e.g., enzymes) [7]. The increase of protein concentration is also connected with the ability of bacteria to exopolysaccharides synthesis [34]. Among others, the acting on membrane integrity by essential oils was also described by Dioa et al. [35]. They investigated the mode of antibacterial action of fennel (*Foeniculum vulgare* Mill.) essential oil. The authors stated that protein release is an efficiency marker confirming the effectiveness of antibacterial activity of essential oils.

## 4. Conclusions

Our investigation demonstrated that plant essential oils can effectively inhibit the growth of saprophytic bacteria *P. orientalis.* Both types of oils, commercial ones and the ones produced in our laboratory, showed a bacteriostatic effect in a low concentration, designated as MIC. What is important is the oils distilled in our laboratory had a bacteriostatic effect at a lower concentration and their antibacterial action was more effective. This effect was proven at individual stages of our work. The issue in question is important for the food industry where the doses of oils should be as low as possible, and should not negatively affect organoleptic properties of products. The lack of negative influence on organoleptic parameters were conduct in the sensory test of salmon in pickle containing individual laboratory EOs in MIC concentrations (data not published). Moreover, oils do not exhibit cyto- and genotoxic characteristics. Of course, before introducing the oil into the model food product and to confirm the above observations, we plan to conduct further studies consisting in analyzing the composition and antimicrobial activity of subsequent oils from the same plants, but obtained from at least five different producers.

## Figures and Tables

**Figure 1 foods-09-00141-f001:**
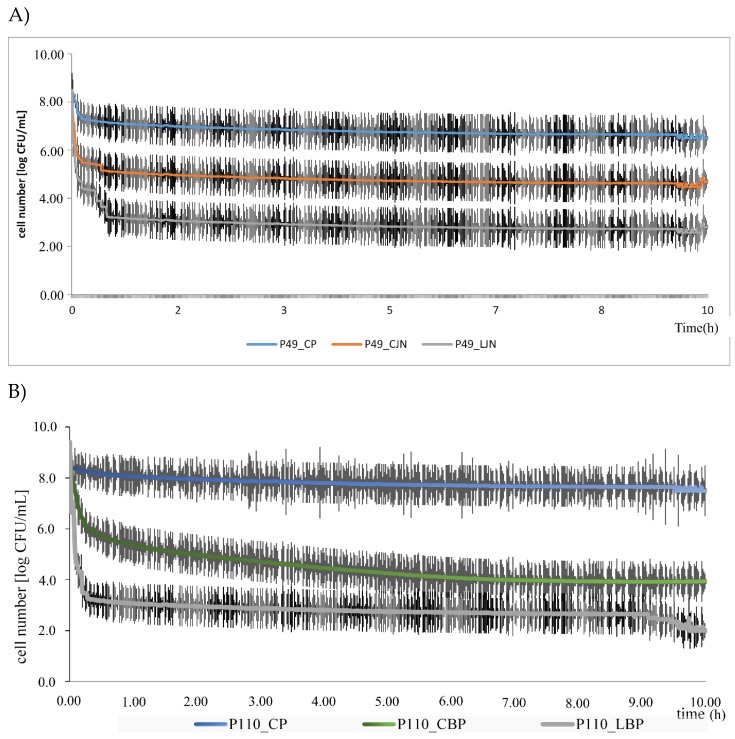
Bacteria growth curve with and without essential oils in Minimal Inhibitory Concentration (MIC) levels (**A**) P49 strain, juniper oil; (**B**) P110 black pepper oil; CP—control probe, CJN—commercial juniper oil, LJN—laboratory juniper oil, CBP—commercial black pepper oil, LBP—laboratory black pepper oil.

**Figure 2 foods-09-00141-f002:**
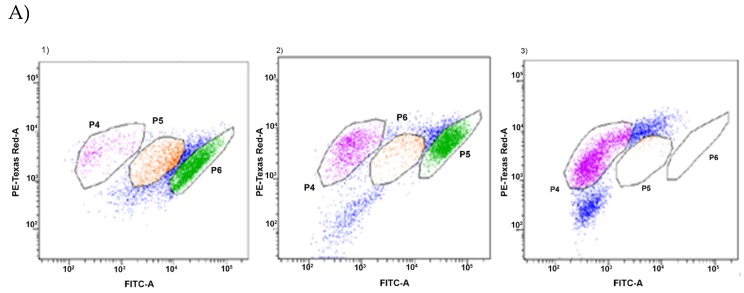
Two-dimensional graph of a distribution of individual fractions in cytometric analyses in case of oils extracted in our laboratory (**A**) and commercial ones (**B**) (1—CP; 2—BP oil; 3—LG oil). Analyses were carried out with *P. orientalis* P49 strain.

**Figure 3 foods-09-00141-f003:**
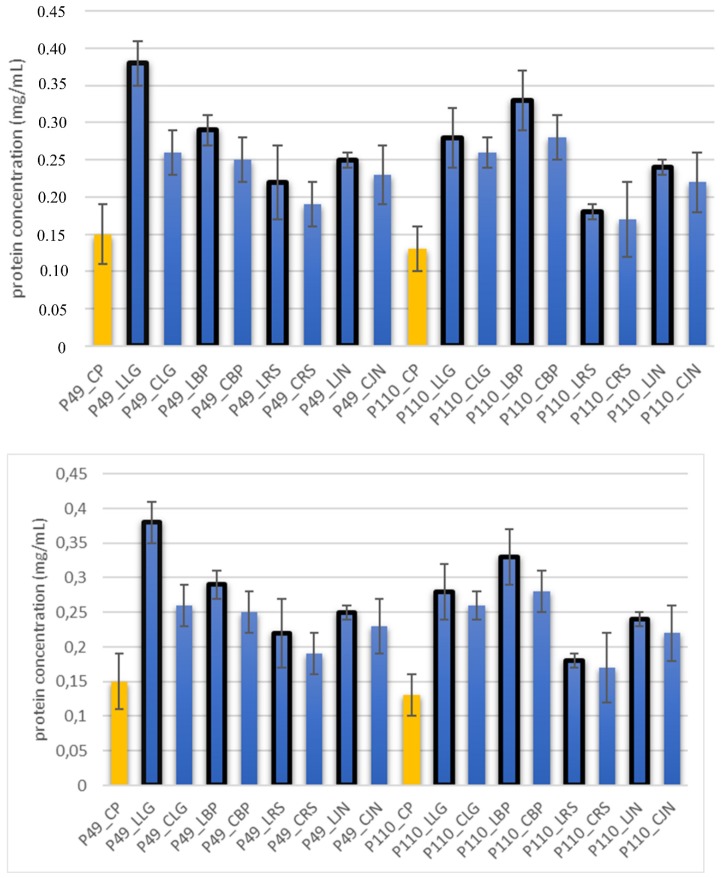
The level of proteins released in bacteria cultivation supplemented with essential oils in MIC levels, control probes are shown as orange columns, columns of laboratory oils have a black frame. CRS—commercial rosemary oil, LRS—laboratory rosemary oil, CBP—commercial black pepper oil, LBP—laboratory black pepper oil, CLG—commercial lemongrass oil, LLG—laboratory lemongrass oil, CJN—commercial juniper oil, LJN—laboratory juniper oil.

**Table 1 foods-09-00141-t001:** Production capacity of essential oil of plant material and a list of cost of obtaining 10 mL of oils (on the Polish market in July 2019; the currency exchange rate as of 12 June 2019).

EO	Commercial	Eco-Farm		
	Price (10 mL of Essential Oil)	Price (100 g of Plant Material)	Price (Plant Material For Obtaining 10 mL of Oil)	Efficiency of Distillation (%)
BP	13.36 EUR (57 PLN)	0.82 EUR 3.50 PLN	13.13 EUR (56 PLN)	1.25
RS	3.28 EUR (14 PLN)	0.70 EUR (3.00 PLN)	9.38 EUR (40 PLN)	0.95
LG	4.45 EUR (19 PLN)	(0.70 EUR) 3.00 PLN	12.66 EUR (54 PLN)	0.18
JN	4.22 EUR (18 PLN)	0.82 EUR 3.50 PLN	5.63 EUR (24 PLN)	1.46

BP—black pepper, RS—rosemary, LG—lemongrass, JN—juniper.

**Table 2 foods-09-00141-t002:** The bioactive composition of both types of oils.

Bioactive Compound/ Oil Type	RI-SLB-5	CRS (%)	LRS (%)	CBP (%)	LBP (%)	CLG (%)	LLG (%)	CJN (%)	LJN (%)
α-thujene	938	nd	0.5	nd	1.4	nd	nd	nd	1.9
α-pinene	939	18.0	18.4	8.0	9.1	nd	nd	1.0	33.1
camphene	953	2.0	2.8	nd	0.8	nd	0.5	nd	0.8
sabinene	974	nd	4.3	10.8	nd	nd	0.5	nd	5.21
ß-pinene	980	8.0	1.6	7.0	9.0	nd	nd	18.0	12.2
ß-myrcene	990	1.5	2.9	nd	2.3	nd	7.4	1.0	11.8
α-phellandrene	1006	nd	nd	nd	4.6	nd	nd	nd	1.13
α-terpinene	1015	1.0	0.5	nd	15.8	nd	nd	nd	5.5
p-cymene	1026	1.0	0.8	nd	0.8	nd	nd	nd	1.1
limonene	1030	2.5	3.0	10.0	14.8	0.2	0.4	nd	9.1
1.8-cineole (eucalyptol)	1031	18	44.2	nd	nd	nd	4.3	nd	nd
β-phellandrene	1042	nd	nd	nd	2.9	nd	nd	nd	1.7
γ- terpinene	1072	nd	0.8	nd	0.5	nd	nd	18.0	1.0
α-terpinolene	1083	nd	nd	nd	0.8	nd	nd	nd	0.9
linalool	1100	nd	0.8	nd	0.8	nd	1.2	nd	nd
camphor	1121	20.0	9.5	nd	nd	nd	4.5	nd	nd
verbenone	1145	0.7	nd	nd	nd	nd	nd	nd	nd
borneol	1156	3.0	1.9	nd	nd	nd	4.5	nd	nd
terpinen-4-ol	1180	nd	1.1	nd	0.5	nd	3.0	nd	1.1
decanal	1206	nd	nd	nd	nd	1.5	nd	nd	nd
neral	1215	nd	nd	nd	nd	28.0	29.7	nd	nd
geranial	1222	nd	nd	nd	nd	35.0	38.5	nd	nd
linalyl acetate	1246	nd	nd	22.0	nd	nd	nd	nd	nd
citral	1268	nd	nd	nd	nd	67.0	31.2	4.3	nd
bornyl acetate	1285	0.5	nd	nd	nd	nd	nd	nd	nd
ß-sinensal	1295	nd	nd	0.5	nd	nd	nd	nd	nd
β-elemene	1345	nd	nd	nd	1.2	nd	nd	nd	0.6
β-cubebene	1348	nd	nd	nd	1.2	nd	nd	nd	0.6
eugenol	1357	nd	nd	nd	nd	nd	nd	nd	nd
geranyl acetate	1384	nd	nd	nd	nd	0.5	1.2	nd	nd
ß-caryophyllene	1442	nd	0.9	20.0	18.8	nd	0.7	nd	8.5
α-muurolene	1498	nd	nd	nd	1.1	nd	0.6	nd	1.9
β-bisabolene	1508	nd	nd	nd	nd	nd	nd	nd	nd
delta-3-Carene	1518	nd	nd	9.0	nd	nd	nd	nd	nd
δ-cadinene	1520	nd	nd	nd	1.5	nd	nd	nd	0.8
myristicin	1546	nd	nd	nd	nd	nd	nd	nd	nd

CRS—commercial rosemary oil, LRS—laboratory rosemary oil, CBP—commercial black pepper oil, LBP—laboratory black pepper oil, CLG—commercial lemongrass oil, LLG—laboratory lemongrass oil, CJN—commercial juniper oil, LJN—laboratory juniper oil. RI-SLB-5—Retention index on SLB-5 column, calculated from the retention time of the compound and the retention times of adjacent n-alkanes by linear interpolation, nd—not detected.

**Table 3 foods-09-00141-t003:** Comparison of antibacterial activity of commercial oils and oils obtained by the authors.

Bacteria Strain	P49						P110					
	Zone Inhibition (mm)		MIC (mg/mL)		Growth Inhibition (%)		Zone Inhibition (mm)		MIC (mg/mL)		Growth Inhibition (%)	
Oil type	CO	LO	CO	LO	CO	LO	CO	LO	CO	LO	CO	LO
RS	0.4 ^a^ ± 0.2	0.6 ^a^ ± 0.2	80.6 ^a^ ± 2.1	8.85 ^b^ ± 1.2	24.5 ^a^ ± 1.4	46.0 ^b^ ± 1.2	7.0 ^a^ ± 0.1	27.0 ^b^ ± 2.6	80.1 ^a^ ± 6.3	8.12 ^b^ ± 1.1	64.5 ^a^ ± 2.4	79.4 ^b^ ± 3.6
BP	2.0 ^a^ ± 0.2	7.2 ^b^ ± 0.3	62.9 ^a^ ± 2.3	6.9 ^b^ ± 0.8	49.9 ^a^ ± 0.7	57.0 ^b^ ± 2.6	0.6 ^a^ ± 0.1	20.0 ^b^ ± 2.1	78.7 ^a^ ± 2.6	7.60 ^b^ ± 0.6	64.76 ^a^ ± 2.0	88.5 ^b^ ± 3.2
LG	5.1 ^a^ ± 1.1	7.6 ^b^ ± 0.2	65.1 ^a^ ± 1.8	3.15 ^b^ ± 0.2	48.5 ^a^ ± 1.8	58.0 ^b^ ± 3.9	18.1 ^a^ ± 1.9	19.0 ^a^ ± 1.2	70.6 ^a^ ± 0.3	3.98 ^b^ ± 0.5	62.7 ^a^ ± 5.7	82.9 ^b^ ± 2.4
JN	2.0 ^a^ ± 0.3	7.6 ^b^ ± 0.8	81.3 ^a^ ± 3.1	8.9 ^b^ ± 1.4	40.7 ^a^ ± 2.5	88.4 ^b^ ± 2.1	2.1 ^a^ ± 0.4	17.0 ^b^ ± 2.0	76.0 ^a^ ± 8.1	9.15 ^b^ ± 1.2	54.9 ^a^ ± 1.5	64.9 ^b^ ± 1.9

^a,b^, Distinct letters indicate significant difference between both oil types in the same experiment and the same bacteria strain (*p* < 0.5). CO—commercial oil, LO—laboratory oil, RS—rosemary, BP—black pepper, LG—lemongrass, JN—juniper.

**Table 4 foods-09-00141-t004:** The comparison of percentage of cells incubated in the presence of commercial oils and of oils produced in our laboratory in fractions with low, medium (viable but nonculturable cells) and high metabolic activity.

Essential Oil	Oils Source	Strain	Low Metabolic Activity (%)	Medium Metabolic Activity (%)	High Metabolic Activity (%)
Control probe		P49	2.3 ^a^ ± 0.3	9.2 ^a^ ± 0.6	80.1 ^c^ ± 0.8
		P110	1.4 ^a^ ± 0.2	4.2 ^a^ ± 0.1	86.1 ^c^ ± 0.6
RS oil	LRS	P49	49.2 ^c^ ± 0.4	19.3 ^b^ ± 0.3	33.2 ^a^ ± 0.4
		P110	33.9 ^b^ ± 0.1	22.1 ^b^ ± 0.2	43.2 ^a^ ± 0.4
	CRS	P49	2.1 ^a^ ± 0.8	5.7 ^a^ ± 0.9	81.8 ^c^ ± 1.2
		P110	3.2 ^a^ ± 0.3	9.5 ^a^ ± 0.3	75.6 ^b^ ± 0.7
BP oil	LBP	P49	53.9 ^c^ ± 0.7	4.9 ^a^ ± 0.6	27.1 ^a^ ± 0.2
		P110	50.9 ^c^ ± 0.3	6.3 ^a^ ± 0.6	29.3 ^a^ ± 0.2
	CBP	P49	2.2 ^a^ ± 0.2	6.8 ^a^ ± 1.2	83.1 ^c^ ± 2.2
		P110	2.0 ^a^ ± 0.1	2.5 ^a^ ± 0.2	91.1 ^c^ ± 0.2
LG oil	LLG	P49	87.4 ^d^ ± 0.2	6.2 ^b^ ± 0.5	1.6 ^d^ ± 0.1
		P110	92.2 ^d^ ± 0.6	4.7 ^b^ ± 0.5	1.2 ^d^ ± 0.1
	CLG	P49	36.7 ^b^ ± 0.5	24.6 ^c^ ± 0.6	25.9 ^a^ ± 0.4
		P110	6.8 ^a^ ± 0.5	5.8 ^a^ ± 0.4	78.3 ^b^ ± 0.5
JN oil	LJN	P49	36.4 ^b^ ± 0.7	27.2 ^c^ ± 0.9	31.9 ^a^ ± 1.4
		P110	39.1 ^a^ ± 0.5	26.6 ^a^ ± 0.3	30.7 ^c^ ± 1.8
	CJN	P49	4.2 ^a^ ± 0.1	18.1 ^b^ ± 2.6	62.8 ^b^ ± 3.3
		P110	2.7 ^a^ ± 0.1	6.0 ^a^ ± 0.6	82.4 ^c^ ± 1.1

^a–d^, Distinct letters within the same column indicate significant difference in the same experiment and the same bacteria strain (*p* < 0.5); CRS—commercial rosemary oil, LRS—laboratory rosemary oil, CBP—commercial black pepper oil, LBP—laboratory black pepper oil, CLG—commercial lemongrass oil, LLG—laboratory lemongrass oil, CJN—commercial juniper oil, LJN—laboratory juniper oil.

**Table 5 foods-09-00141-t005:** Cells’ morphological changes under the influence of essential oils. CRS—commercial rosemary oil, LRS—laboratory rosemary oil, CBP—commercial black pepper oil, LBP—laboratory black pepper oil, CLG—commercial lemongrass oil, LLG—laboratory lemongrass oil, CJN—commercial juniper oil, LJN—laboratory juniper oil.

	Control Probe		LRS		CRS		LBP		CBP		LLG		CLG		LJN		CJN	
	Length (μm)	Width (μm)	Length (μm)	Width (μm)	Length (μm)	Width (μm)	Length (μm)	Width (μm)	Length (μm)	Width (μm)	Length (μm)	Width (μm)	Length (μm)	Width (μm)	Length (μm)	Width (μm)	Length (μm)	Width (μm)
**P49**	1.02 ± 0.3	0.49 ± 0.1	0.82 ± 0.2	0.73 ± 0.11	0.71 ± 0.1	0.72 ± 0.2	0.78 ± 0.3	0.82 ± 0.2	0.85 ± 0.2	0.83 ± 0.1	0.71 ± 0.1	0.69 ± 0.1	0.76 ± 0.1	0.73 ± 0.1	0.65 ± 0.2	0.67 ± 0.2	0.73 ± 0.3	0.75 ± 0.2
	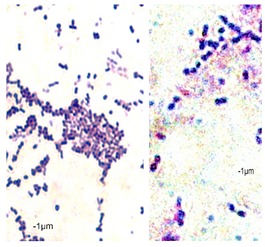				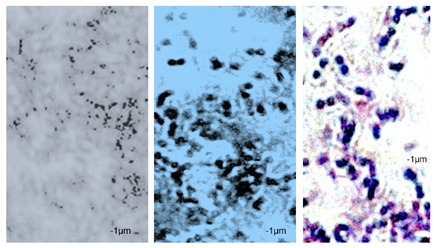						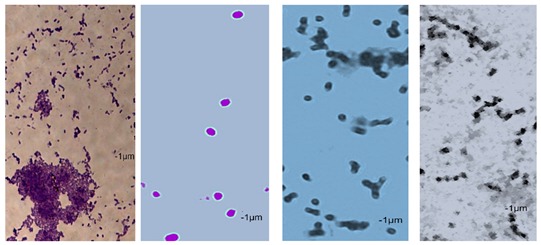							
**P110**	1.21 ± 0.1	0.62 ± 0.1	0.69 ± 0.1	0.67 ± 0.17	0.65 ± 0.1	0.68 ± 0.1	0.71 ± 0.2	0.74 ± 0.1	0.84 ± 0.2	0.81 ± 0.1	0.80 ± 0.2	0.76 ± 0.2	0.82 ± 0.3	0.84 ± 0.1	0.75 ± 0.1	0.73 ± 0.2	0.80 ± 0.2	0.78 ± 0.2
	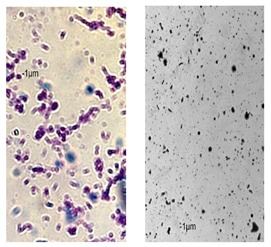										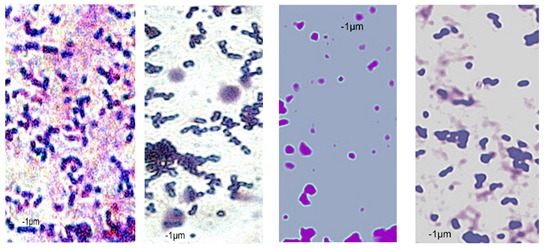							

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
