# Peer review of "Comparative Evaluation of Piper nigrum, Rosmarinus officinalis, Cymbopogon citratus and Juniperus communis L. Essential Oils of Different Origin as Functional Antimicrobials in Foods"

_foods, 2020, doi:10.3390/foods9020141_

Round 1
Reviewer 1 Report
The authors added more information in the introduction part. However, they didn't mention what exactly is the "low concentration" of EOs as food preservatives. How do the MIC they determined compared to the actual EOs concentration in food?
What's more, the sample capacity is still too small so that the conclusion is not convincing. If the authors are trying to avoid doing extra experiments, they need to make great efforts to polish their manuscript.
The English still need a lot improvement.
Author Response
The authors added more information in the introduction part. However, they didn't mention what exactly is the "low concentration" of EOs as food preservatives. How do the MIC they determined compared to the actual EOs concentration in food?The explaining what low concentration is, is a very important issue, thank you for your valuable advice. The Introduction section has been supplemented by that information. Moreover, some examples of MIC level has been added. It has been also underlined that the MIC values determined in vitro are lower that MIC values in foods.
What's more, the sample capacity is still too small so that the conclusion is not convincing. If the authors are trying to avoid doing extra experiments, they need to make great efforts to polish their manuscript.We have clearly indicated in the conclusions that our research must be continued. To confirm the results and thus the final observations, before introducing oils into the model product, we intend to carry out further analyzes of oils from the same plants, but obtained from at least 5 further producers.
English still need a lot improvement.I send my article for English editing in FOODS.
Reviewer 2 Report
Even if the quality of figures could be improved, your replies to my comments are satisfying
Author Response
Even if the quality of figures could be improved, your replies to my comments are satisfyingI did everything possible to improve the quality of the tables and figures. This is certainly strongly visible compared to the previous version of the article. The diagrams have been developed in the graphic programs of the microscope, cytometer and spectrophotometer. Here the quality is dictated by the equipment. In my previous work, published in FOODS, there are analogous graphics (although the graphics in this work are of even better quality).
This manuscript is a resubmission of an earlier submission. The following is a list of the peer review reports and author responses from that submission.
Round 1
Reviewer 1 Report
Lines 88-90
What’s the Eos concentration in the 50 µL placed on the plate? Is it comparable with the 25 μg/disc of gentamicin?
Line 99
The article Prudent et al., 1995 lacks in the reference list
Line 102
.. ranging (determined based on 101 previous observations) from 6.0 to 96.0 mg/ml … May be it is due to my scarce experience in microbiological tests but I don’t understand why authors use different units, 50 µL/disc 25 μg/disc above and here mg/ml; how many mg/disk? In my opinion, taking into account that data must be referred to cells number (CFU/mL), the concentration would be the better choice but why in lines 88-90 other units are used?
Results and discussion
I am unable to understand the difference between the commercial and “obtained in lab” EOs. There is any difference in the distillation method or only differ for the origin of the raw materials?
Table 1
It would be better to list all prices in Euro
Lines 164-165
The period “To compare the content of bioactive components, a GC/MS analysis was performed. The results are summarized in Table 2.” must be moved at the end of line 168
Table 2
The second column is not explained at all. Reading the text it is clear that nd means “not detected” but it also could mean not determined; so, it would be better to add the meaning in the note. Really it would be better to write “< LOD” and to list the single LOD of each compound in an external file if foreseen.
In the Introduction, authors correctly state ” Because they are a natural plant product, their chemical composition can vary depending on climatic conditions where the plant was cultivated, and is different in various parts of the plant from which the oil was obtained.” At what extent? The comparison between the commercial and “in lab” EOs is really significant?
More than 30 compounds are listed; “Such differences in both oils’ composition certainly influenced the effectiveness of antibacterial action of essential oils, because every single compound has a strictly defined role in antimicrobial activity.” have all an antibacterial activity at the found %? For each one, authors obtained a calibration curve? At least one chromatogram must be reported in order to evaluate the analysis time and, overall, the resolution between the peaks.
Table 3
Suffer of formatting problems. The two parts relative to the different bacteria would be better separates, as an example by a vertical line. [mg d.m./mL]? Number must be reported with the same number of decimals. For the last two column p<0 is not true. Even if declared in the experimental part, in the caption must be specified that ± x is not a Standard Deviation but the difference between the value of duplicates tests. The same must be done for Tables 4-5 and for the error bars in the histogram of fig 3. I wonder that at big differences between “zone inhibition” correspond a relatively low differences between “Growth inhibition”. Just as an example data for BP on P110: at 0.6 mm of zone inhibition corresponds about 65% of growth inhibition that differ only of about 23% from a 20 mm zone inhibition; more, such difference is just a little higher than the one obtained for LG on the same bacteria where no significant difference results for the zone inhibition.
Fig. 2
The graphic quality is very scarce
Table 5
No labels are present in the single images; they are in the order listed in the caption and the first is the control?
Conclusion
The paper is very rich of data but my main perplexity regards the real need of the done comparison. In my opinion, if EOs are proposed as antimicrobial for food, they must be compared with compounds actually used in food at that aim.
Authors correctly stated “.. should not negatively affect organoleptic properties of products.” On such bases the perception threshold of each compound should be reported (may be that they are reported in literature) and compared with the MIC.
Reviewer 2 Report
In this manuscript, Komosa et al tested the effectiveness of EOs from pepper, rosemary, lemongrass and juniper extracted in laboratory with commercial EOs. They conducted GC to analyze the active ingredients content, and compared the antibacterial activity. They suggested that both types of oils can effectively inhibit the growth of saprophytic bacteria P. orientalis by targeting bacterial cell wall and cell membrane.
The manuscript is quite poorly written.
The manuscript has numerous grammatical, word choice errors, which makes it really hard to interpret (such as in line 40, the wrong subject of the sentence, wrong space between words, and some abbreviations like P. orientalis in abstract need to be stated as full name, etc). It would require extensive editing for appropriate English usage.
The introduction needs to be more knowledgeable.
Except listing the literatures, the authors are suggested to give examples of the antimicrobial effect of commercial EOs and their mode of action (line 47-49). And it’s necessary to state why the authors choose these four plants for EOs comparison.
The amount and quality of the work are not sound enough to support the conclusion.
First, the authors compared the effectiveness of commercial and laboratory EOs from four plants. However, there will be huge differences between different brands of commercial products, even different batches of products. For the laboratory extracts, there will also be huge differences between the samples from different origins, just as the authors stated: “Because they are a natural plant product, their chemical composition can vary depending on climatic conditions where the plant was cultivated, and is different in various parts of the plant from which the oil was obtained.” Exactly. Only comparing one commercial product and one laboratory extract could not lead to any conclusion.
Second, it’s not meaningful to compare the commercial product with laboratory extract if the corresponding ingredients are not clearly known. The EOs extracted from plants are freshly made with elaborate laboratory technique, while the commercial products are under industrial production, which some of the active ingredients could be lost, or degraded over time. Thus even they are comparable, the conclusion is not applicable to practical use.
The authors are suggested to compare more commercial products and more samples from different areas, and discuss all the factors which could affect the effectiveness of EOs in discussion.